# Rac1 GTPase activates the WAVE regulatory complex through two distinct binding sites

Baoyu Chen[1,2†‡*], Hui-Ting Chou[3†§], Chad A Brautigam[1,4], Wenmin Xing[1,2], Sheng Yang[5], Lisa Henry[1,2], Lynda K Doolittle[1,2], Thomas Walz[6], Michael K Rosen[1,2*]

[1]Department of Biophysics, University of Texas Southwestern Medical Center at Dallas, Dallas, United States; [2]Howard Hughes Medical Institute, University of Texas Southwestern Medical Center at Dallas, Dallas, United States; [3]Department of Cell Biology, Harvard Medical School, Boston, United States; [4]Department of Microbiology, University of Texas Southwestern Medical Center, Dallas, United States; [5]Roy J. Carver Department of Biochemistry, Biophysics and Molecular Biology, Iowa State University, Ames, United States; [6]Rockefeller University, New York, United States

*For correspondence:
stone@iastate.edu (BC);
michael.rosen@utsouthwestern.
edu (MKR)

[†]These authors contributed
equally to this work

Present address: [‡]Roy J. Carver
Department of Biochemistry,
Biophysics and Molecular
Biology, Iowa State University,
Ames, United States; [§]CryoEM
Shared Resources, Janelia
Research Campus, Howard
Hughes Medical Institute,
Ashburn, United States

Competing interests: The
authors declare that no
competing interests exist.

Reviewing editor: Axel T
Brunger, Stanford University
Medical Center, United States

**Abstract** The Rho GTPase Rac1 activates the WAVE regulatory complex (WRC) to drive Arp2/3 complex-mediated actin polymerization, which underpins diverse cellular processes. Here we report the structure of a WRC-Rac1 complex determined by cryo-electron microscopy. Surprisingly, Rac1 is not located at the binding site on the Sra1 subunit of the WRC previously identified by mutagenesis and biochemical data. Rather, it binds to a distinct, conserved site on the opposite end of Sra1. Biophysical and biochemical data on WRC mutants confirm that Rac1 binds to both sites, with the newly identified site having higher affinity and both sites required for WRC activation. Our data reveal that the WRC is activated by simultaneous engagement of two Rac1 molecules, suggesting a mechanism by which cells may sense the density of active Rac1 at membranes to precisely control actin assembly.

DOI: https://doi.org/10.7554/eLife.29795.001

## Introduction

Dynamic rearrangements of the actin cytoskeleton play central roles in cellular processes, ranging from cell migration and adhesion to endocytosis and intracellular vesicle trafficking (*Skau and Waterman, 2015*). In many of these processes, actin dynamics are spatially and temporally controlled by members of the Wiskott-Aldrich Syndrome Protein (WASP) family. These proteins integrate a diverse array of upstream signals and transmit them through their conserved VCA sequence to the Arp2/3 complex, which, in turn, nucleates actin filaments to create branched actin networks at membranes (*Campellone and Welch, 2010*; *Padrick and Rosen, 2010*). In most WASP family proteins, the VCA element is inhibited in the resting state. Inhibition is mediated either by autoinhibition within a single polypeptide chain, as in WASP and N-WASP, or by trans-inhibition within large multi-protein assemblies, as in WAVE and WASH (*Rotty et al., 2013*). Multiple signals, including binding to ligands (e.g. Rho family GTPases, phosphoinositide lipids and membrane receptors) and covalent modifications (e.g. phosphorylation and ubiquitination), often act cooperatively to relieve inhibition and concomitantly recruit WASP proteins to their sites of action at membranes (*Chen et al., 2014a*, *2010*; *Hao et al., 2013*; *Lebensohn and Kirschner, 2009*; *Padrick and Rosen, 2010*; *Prehoda et al., 2000*; *Torres and Rosen, 2003*; *2006*).

**eLife digest** Our cells contain a network of filaments made up of a protein called actin. Just like the skeleton that supports our body, the actin 'cytoskeleton' gives a cell its shape and strength. Actin filaments are also critical for many other processes including enabling cells to move and divide. The assembly of actin filaments must be properly controlled so that they are formed at the right time and place within the cell.

A complex of proteins known as the WAVE Regulatory Complex (WRC) promotes the assembly of actin filaments. The complex contains a region called the VCA, which is able to bind to and activate another protein to make the new actin filaments. The WRC regulates filament assembly by controlling the availability of the VCA in a way that is similar to opening and closing a safe box. When new actin filaments are not needed, the safe box is closed and the VCA is not available. However, when cells need to make new actin filaments, the WRC is opened to release the VCA region so that it is able to bind to the filament-producing protein.

Previous studies have shown that a protein called Rac1 acts as a key to open the WRC and trigger actin filament assembly. But it remains unclear how this works. A major obstacle to studying this process is that Rac1 and the WRC only weakly interact with each other, which makes it difficult to capture the interaction under experimental conditions.

To overcome this obstacle, Chen et al. tethered a Rac1 molecule to the WRC in order to make the interaction more stable. A technique called cryo-electron microscopy was used to study the three-dimensional shape of this Rac1-WRC complex. Unexpectedly, Rac1 was attached to a different part of the WRC than the site predicted by previous studies. Further experiments showed that Rac1 needs to bind to both of these sites at the same time in order to open the WRC and release VCA, similar to using two keys to open one safe box for increased security.

Some cancers, neurological disorders and other diseases can be caused by defects in WRC and Rac1 activity. Therefore, these findings could lead to new ways to treat these conditions in human patients.

DOI: https://doi.org/10.7554/eLife.29795.002

The WASP family members WAVE1, WAVE2 and WAVE3 are essential to actin dynamics needed for normal development and function of most eukaryotic organisms, and are also implicated in many cancers (*Bisi et al., 2013*; *Lane et al., 2014*; *Takenawa and Suetsugu, 2007*; *Yanagisawa et al., 2013*). In cells, each WAVE protein functions exclusively within a 400 kDa, heteropentameric assembly, termed the WAVE Regulatory Complex (WRC), which also contains the proteins Sra1, Nap1, Abi2, and HSPC300 (or their homologs) (*Stovold et al., 2005*). Previously, we reported crystal structures of an inhibited WRC assembly containing all five subunits, but lacking the disordered C terminus and SH3 domain of Abi2 and the proline-rich region of WAVE1 (mini-WRC) (*Chen et al., 2010*), and of this assembly bound to a WRC Interacting Receptor Sequence (WIRS) motif peptide derived from the adhesion receptor, protocadherin 10 (*Chen et al., 2014a*). The structure revealed that Sra1 and Nap1 form an elongated dimer of about $100 \times 100 \times 200$ Å. WAVE1, Abi2 and HSPC300 form a trimer as an elongated 4-helix bundle, which aligns along the long axis of the Sra1/Nap1 dimer (*Figure 1A*). Immediately following the helix bundle, an extended sequence of ~90 a.a. from WAVE1 'meanders' across the surface of Sra1 as a loose collection of loops and helices (the meander region). The VCA element of WAVE1 is sequestered at the VCA-binding site by multiple interactions between its V and C helices and a group of helices from both Sra1 and the meander region of WAVE1 itself, explaining how WAVE1 is inhibited within the WRC (*Chen et al., 2010*).

The structures also suggested how the WRC might be activated and recruited to membranes by the combined actions of the Rho GTPase Rac1, the acidic phospholipid $PIP_3$, phosphorylation on both Ser and Tyr residues, and membrane receptors containing WIRS motifs. Among these, the GTPase Rac1 functions as a canonical activator, whose direct interaction with the WRC is necessary, and in some reports, sufficient to drive activation (*Chen et al., 2014a*; *Chen et al., 2010*; *Eden et al., 2002*; *Ismail et al., 2009*). Based on mutagenesis and biochemical data, we previously proposed that Rac1 binds to a conserved surface at one end of Sra1 adjacent to the VCA-binding site (referred to as the 'A Site' in *Figure 1B*). We hypothesized that binding of Rac1 could cause

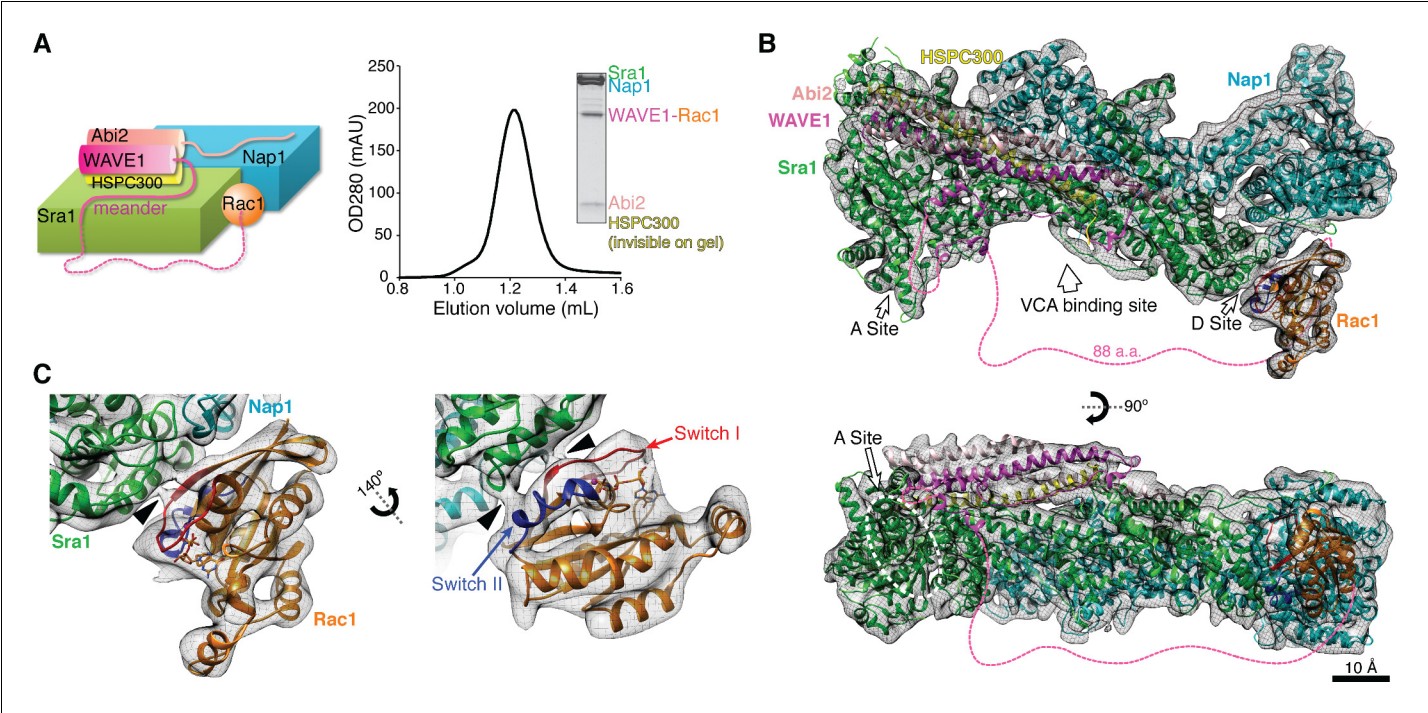

**Figure 1.** Cryo-EM structure of the WRC-Rac1 complex. (**A**) Schematic representation (left) and size-exclusion chromatography profile (right) of the ΔWRC230-Rac1 complex, with Coomassie blue-stained SDS-PAGE gel of the peak fraction shown. Dashed line indicates the tethering sequence between WAVE1 meander region and Rac1. (**B**) Overall EM density and the fitted model of the ΔWRC230-Rac1 complex. WAVE1(130–147, 161–230) and (GGS)$_6$ leading to Rac1 have no EM density and are shown as dotted lines. (**C**) Close-up views of Rac1 fitted into the extra density not accounted for by the WRC. Black arrow heads indicate close contacts between Rac1 and the WRC.

DOI: https://doi.org/10.7554/eLife.29795.003

The following figure supplements are available for figure 1:

**Figure supplement 1.** A covalent ΔWRC230-Rac1 complex.
DOI: https://doi.org/10.7554/eLife.29795.004

**Figure supplement 2.** Representative EM images as well as class averages and 3D maps of the ΔWRC230-Rac1 complex.
DOI: https://doi.org/10.7554/eLife.29795.005

**Figure supplement 3.** Cryo-EM density map for the ΔWRC230-Rac1 complex.
DOI: https://doi.org/10.7554/eLife.29795.006

structural changes that propagate through the meander region to the VCA to promote VCA release and consequent activation toward the Arp2/3 complex (*Chen et al., 2010*). However, in the absence of a WRC-Rac1 complex structure, it remained uncertain exactly how Rac1 binds to and activates the WRC.

Here we report the structure of a WRC-Rac1 complex at 7 Å resolution determined by single-particle cryo-electron microscopy (cryo-EM), together with complementary mutagenesis and biochemical analyses of the complex. Surprisingly, Rac1 is not found on the binding site we predicted previously. Rather, it binds to a conserved site on the opposite end of Sra1. Our biochemical data, including quantitative pull-down, membrane recruitment, and analytical ultracentrifugation (AUC) assays with WRC mutants, confirm that both the original site and the new site can bind Rac1, the latter with approximately 40-fold higher affinity. Actin assembly experiments further reveal that both sites contribute to activation of the WRC toward the Arp2/3 complex. These data lead to a model in which maximal WRC activation requires simultaneous engagement of two Rac1 molecules on the membrane. This bivalent activation model suggests a mechanism by which cells may precisely control actin dynamics at membranes by sensing the density of active Rac1 produced by upstream signaling pathways.

## Results

### Engineering a stable WRC-Rac1 complex

Initially we sought to determine the crystal structure of an intermolecular complex of the WRC bound to Rac1. Despite extensive efforts, however, we were unable to obtain diffracting crystals of such an assembly. These investigations led us to an engineered WRC-Rac1 complex with improved stability, which proved suitable for structure determination by cryo-EM (*Figure 1A,B*). Compared to the mini-WRC construct previously used for crystallography (*Chen et al., 2010*), we incorporated the following modifications to stabilize Rac1 binding (*Figure 1A*). First, we removed the VCA sequence from the WRC (herein named ΔWRC), a modification that we previously showed increases the affinity for Rac1 ~10 fold (*Chen et al., 2010*). Second, we genetically fused Rac1 to the C terminus of WAVE1 (a.a. 1–230) through a flexible linker (GGS)$_6$. The previous crystal structures (and the cryo-EM structure reported here) showed that WAVE1 is only ordered to residue L184 within the WRC, affording the system here an effective linker of 65 amino acids. This linker,~230 Å if fully extended, would provide ample length to connect Rac1 to the previously demonstrated binding site, which is approximately 70 Å from residue L184 of WAVE1. Based on experiments measuring binding to immobilized GST-Rac1, the tethered Rac1 protected the WRC from binding to GST-Rac1 much more efficiently than the un-tethered Rac1 protected the wild type WRC, suggesting the tethered Rac1 binds the WRC in a physiologically relevant manner (*Figure 1—figure supplement 1A*). Third, we introduced two point mutations, Q61L and P29S, into the fused Rac1. The Q61L mutation abolishes the GTP hydrolysis activity of Rac1 (*Der et al., 1986*), maintaining the protein in the GTP-bound state. The P29S mutation was identified as one of the major somatic mutations in melanoma (*Hodis et al., 2012*; *Krauthammer et al., 2012*), and was shown to increase the affinity of Rac1 for the effectors PAK1 (p21 protein activated kinase 1) and MLK3 (mixed-lineage kinase 3) (*Hodis et al., 2012*; *Krauthammer et al., 2012*). Similarly, we found that the P29S mutation also increases the affinity of Rac1 for the WRC (*Figure 1—figure supplement 1B and C*). Together, these optimizations facilitated stable association of Rac1 with the WRC. To distinguish this construct from the mini-WRC used for crystallography, we name it the ΔWRC230-Rac1 complex.

### Cryo-EM Structure of the ΔWRC230-Rac1 Complex

We first examined the ΔWRC230-Rac1 complex by electron microscopy of negatively stained specimens, which showed images of monodisperse particles of homogeneous size and shape (*Figure 1—figure supplement 2A*). This prompted us to collect cryo-EM images of vitrified samples for structure determination (*Figure 1—figure supplement 2B*). We first manually picked ~12,000 particles from 489 images and calculated class averages using the iterative stable alignment and clustering (ISAC) procedure (*Yang et al., 2012*). Some of the resulting averages showed extra density in addition to the WRC, indicating the presence of Rac1 (*Figure 1—figure supplement 2C*, arrows). We further collected 1684 images, from which a total of 160,591 particles were automatically picked, and then processed using Relion (*Scheres, 2012*). The pooled particle images were aligned to the low-pass filtered crystal structure of the mini-WRC (PDB 3P8C) (*Chen et al., 2010*) and subjected to 3D classification (*Figure 1—figure supplement 2D*). After further refinement, we obtained a final density map from 29,784 particles at a resolution of 7 Å (*Figure 1—figure supplement 3*).

The overall EM density map clearly revealed a structure containing both the WRC and Rac1 (*Figure 1—figure supplement 3A*), with Rac1 bound to Sra1 at one end of the WRC (detailed below). The mini-WRC crystal structure (PDB 3P8C) could be unambiguously docked into the EM density map, with the majority of the secondary structural elements fitting into the density without need for adjustment (*Figure 1B*). Several surface loops in Sra1 and Nap1 that were not observed in the crystal structure also have defined EM densities (*Figure 2A*). The EM map shows no density for the V and C helices of the VCA region of WAVE1 (*Figure 2B*, right panel, elements colored black), which was present in the mini-WRC crystal structure but was removed in the ΔWRC230-Rac1 complex used here. No density was observed for the first helix (α1) of Sra1 (*Figure 2B*, left panel). In crystals of mini-WRC this helix did not bind in intra-complex fashion, but rather to an adjacent WRC in the crystal lattice. Density was also not observed for residues 161–230 of WAVE1 or the (GGS)$_6$ linker (*Figure 2B*, right panel), which leads to the N terminus of the fused Rac1, suggesting that this sequence of 88 residues does not form a stable structure in the complex. No major conformational

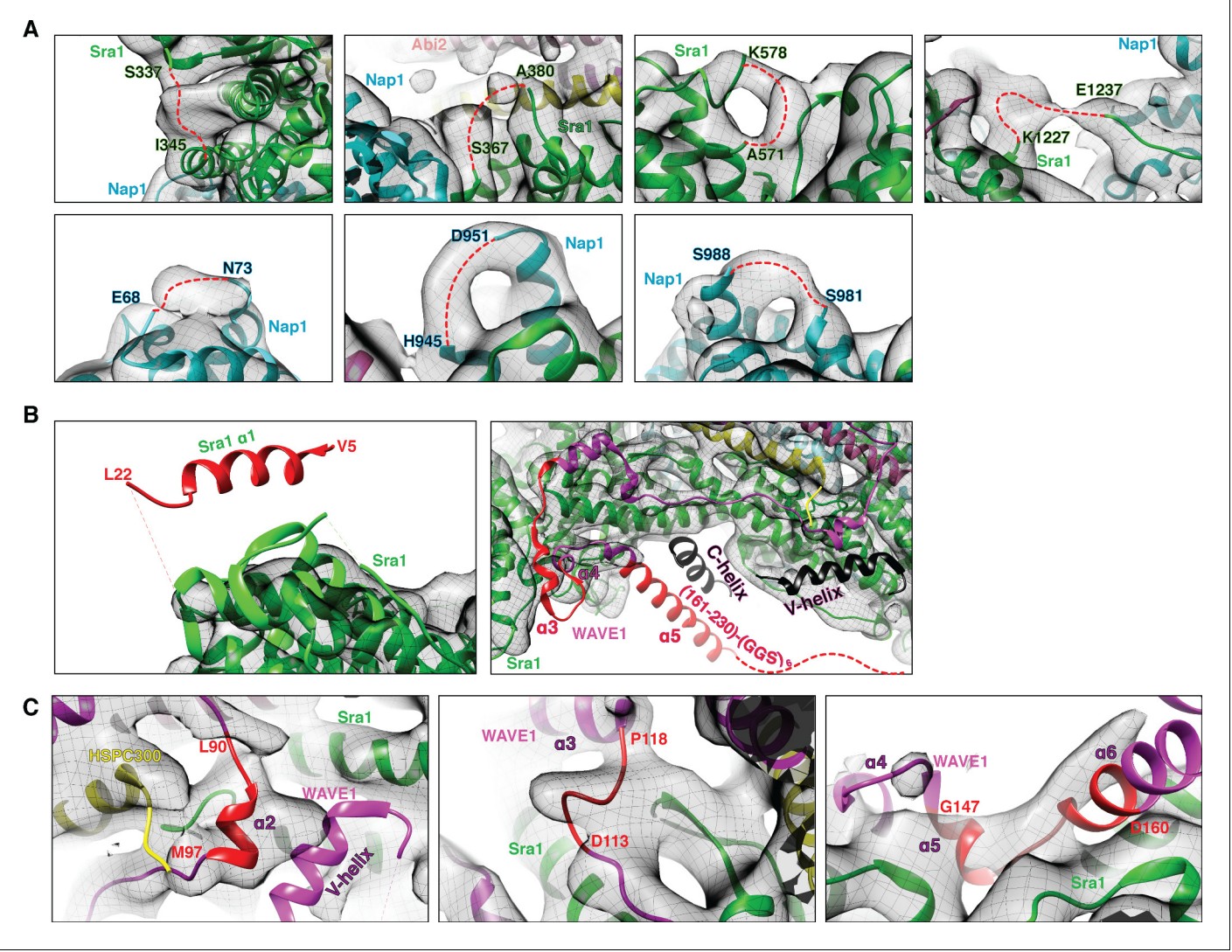

**Figure 2.** Fitting of the WRC crystal structure to the EM density. (**A**) EM density for surface loops (represented by red dashed lines) missing in the mini-WRC crystal structure. (**B**) Elements observed in the crystal structure but lacking densities in the EM reconstruction, with elements present in the ΔWRC230-Rac1 construct in red and elements absent from the ΔWRC230-Rac1 construct in black. (**C**) Several structural elements (red) in the meander sequence have EM densities suggestive of different orientations.

DOI: https://doi.org/10.7554/eLife.29795.007

changes of the WRC were observed within the limits of our 7 Å resolution, except for the meander region of WAVE1 (a.a. 90–184) (*Figure 2C*), which formed a series of helices (α2-α6) arrayed across the surface of Sra1 in the mini-WRC crystal structure (*Chen et al., 2010*). In the EM map, the meander region generally has weak density. Several regions either adopt an orientation different from the crystal structure (a.a. 90–97 in α2, a.a. 113–118 prior to α3, and a.a. 148–160 between α5 and α6), or have missing density (a.a. 130–147 between α3 and α5 and a.a. 161–182 of the long α6 helix) (*Figure 2C*). In the crystal structure of the inhibited mini-WRC, helices α2 and α6 directly bind the V and C regions of the VCA, respectively. Mutagenesis studies suggest that these interactions act cooperatively to sequester the VCA, inhibiting its activity toward the Arp2/3 complex (*Chen et al., 2010*). The observations here support this idea, since removing the VCA (and possibly Rac1 binding as well) reciprocally appears to destabilize the meander region.

The only major extra density not accounted for by the WRC was located on a surface of Sra1 ~120 Å from the previously proposed Rac1 binding site, on the opposite end of the long axis of the WRC (*Figure 1B*). This density unambiguously accommodated the crystal structure of Rac1

P29S, including all secondary structure elements and most loops (PDB 3SBD) (*Krauthammer et al., 2012*) (*Figure 1C*). In this position, the N terminus of Rac1 is located ~100 Å from the last ordered residue of WAVE1 to which it is linked, a distance that could readily be spanned by the 88 disordered residues of the tether. Note that none of our EM maps resulting from 3D classification indicated that Rac1 was bound to other regions of the WRC. Even supervised classification that included as reference a model of the WRC with Rac1 bound to the previously predicted binding site failed to produce a map that revealed additional density at this site (data not shown).

Rac1 and other small GTPases use two conserved structural elements, Switch I (a.a. 25–39) and Switch II (a.a. 57–75), to bind and activate downstream signaling proteins in a nucleotide-dependent manner (*Mott and Owen, 2015*). In our model of the complex, both Switch I and Switch II of Rac1 make direct contact with the WRC, through a conserved surface on the Sra1 subunit (a.a. 957–979, arrow heads in *Figure 1C*). This observation further validates the EM density map, which was derived without using crystal structures of Rac1 as a reference.

## Rac1 binds to two distinct sites on the WRC with different affinities

In our structure Rac1 is not located at the site predicted by our previous biochemical/mutagenesis analyses. That site is a conserved surface patch on Sra1 near the α4-α6 helices of the WAVE1 meander region, adjacent to the VCA binding site (*Figure 1B*, *Figure 3A*) (*Chen et al., 2010*). Instead, in the EM reconstruction, Rac1 binds to a different conserved surface at the opposite end of the rod-shaped Sra1 subunit, which is distant from the meander-VCA region (*Figure 1B*, *Figure 3A*). For clarity, here we name the previously described site the 'A Site' (for <u>A</u>djacent Site), and the site observed in the EM structure the 'D Site' (for <u>D</u>istant Site).

How can the previous biochemical analyses be reconciled with the location of Rac1 found in the present structure? In our previous work, we found that the A Site was one of the few highly conserved surface patches on the WRC. We showed that mutating any of three surface residues (C179R, R190D, or M632D), or one pair of residues (E434K/F626A), at the A Site disrupted Rac1 binding in qualitative GST pull-down experiments, and that the two mutations analyzed in a quantitative equilibrium dialysis experiment (R190D and E434K/F626A) both decreased affinity (*Chen et al., 2010*). In comparison, the D Site is also conserved, although less so than the A Site (*Supplementary file 1* and *Figure 3—figure supplement 1*). To test if Rac1 also binds to the D Site, we introduced point mutations to its surface residues and examined the mutants in a battery of assays. We divided the surface into six distinct patches and mutated them separately (color coded in *Figure 3B*): P957A/K958D/I959A, R961D/P963A/R964D, Y967A, G971W, E974A/F975A/H978A/Q979A, and D1122A/E1123A. We also examined a point mutation, S969F, which is buried immediately beneath the D Site (labeled in red in *Figure 3C and D*). The equivalent mutation, S968F, in Cyfip2 (the mouse ortholog of Sra1) was shown to cause an altered cocaine response in mice through an unknown mechanism (*Kumar et al., 2013*). Finally, we also reanalyzed a previously tested mutation at the A Site, C179R (labeled in black in *Figure 3A*). As described in the Protein Purification section of the Materials and Methods, all of the WRC mutants appear to be properly folded and assembled.

We first examined the ability of immobilized GST-Rac1 to retain the WRC mutants. GST-Rac1 loaded with GMPPNP (a poorly hydrolyzable GTP analog) could efficiently retain wild type (WT) WRC as well as one of the six D Site mutants, D1122A/E1123A (*Figure 3E*). All other mutations at the D Site substantially diminished binding, suggesting that this region mediates high-affinity interactions with Rac1. The D1122A/E1123A mutation is located at the periphery of the D Site, likely explaining its lack of effect on binding (*Figure 3B*, grey patch). Consistent with our previous report (*Chen et al., 2010*), the A Site mutation, C179R, also decreased binding under the same experimental conditions, albeit more moderately (*Figure 3E*). The observed interactions are specific to active Rac1, since Rac1-GDP did not bind to any of the WRCs (*Figure 3—figure supplement 2A*). The S969F mutation did not appreciably affect binding (*Figure 3E*).

We obtained similar results under different reaction conditions, employing the high-affinity mutant Rac1 (Q61L/P29S) at a higher salt concentration (100 mM NaCl, vs. 50 mM above) (*Figure 3—figure supplement 2B*). Under these conditions, which favor binding to the WRC, the A Site mutation C179R, and the D Site mutations, P957A/K958D/I959A, G971W and E974A/F975A/H978A/Q979A, all weakened binding to a moderate degree, whereas the R961D/P963A/R964D and Y967A mutations at the D Site still eliminated observable binding. The D1122A/E1123A and S969F mutations had no effect. These data suggest that the A Site makes a modest contribution to the total

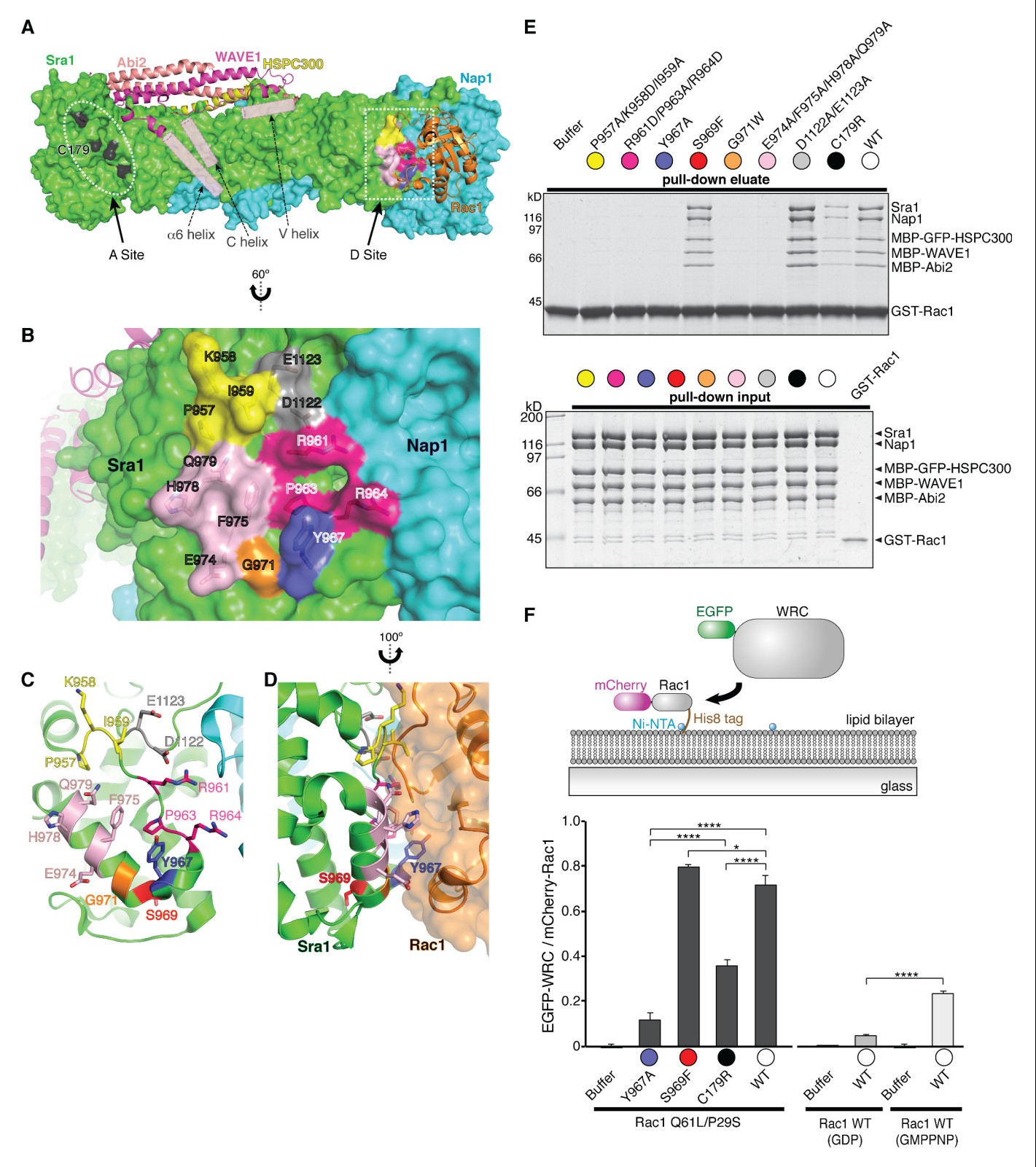

**Figure 3.** Rac1 binds to two distinct sites on the WRC. (**A**) Overall structure of the ΔWRC230-Rac1 complex showing the locations of the A site and D site. Sra1 and Nap1 are shown in surface representation; WAVE1, Abi2, HSPC300 and Rac1 are shown in ribbon representation. Semitransparent pink cylinders are used to refer to the positions of WAVE1 α6 helix (without observable EM density) and the V and C helices (only present in the mini-WRC). Residues in the A Site whose mutation was previously shown to impair Rac1 binding are black. Surface patches in the D Site mutated here are colored

*Figure 3 continued on next page*

*Figure 3 continued*

as in (**B–F**). (**B**) Close-up view of the D Site with semitransparent surface and key side chains shown under the surface. Rac1 was removed for clarity. (**C–D**) top and side views of the D Site, showing the location of the buried residue S969 (red). (**E**) Coomassie blue-stained SDS-PAGE gel showing GST-Rac1-GMPPNP (WT) pull-down assay of WRCs bearing the indicated mutations in Sra1 at the Rac1-binding sites. Top panel shows elution of material retained on the beads, bottom panel shows input control. See non-equilibrium GST pull down assay section of Experimental Methods for details of constructs. (**F**) Top: schematic of membrane recruitment assay using supported lipid bilayers. WRC construct was same as in panel E. Bottom: quantification of the EGFP-WRC/mCherry-Rac1 signals for each reaction used 10 separate TIRF images, with five each collected from two independent experiments. Error bars represent standard error, and p values were calculated by pairwise comparison using Tukey's test (*p<0.05, and ****p<0.00005).

DOI: https://doi.org/10.7554/eLife.29795.008

The following figure supplements are available for figure 3:

**Figure supplement 1.** Surface conservation of the Sra1 subunit in the ΔWRC230-Rac1 complex (Rac1 removed for clarity), with a gradient of color from green to white representing the most conserved surface residues (ConSurf score = 9) to the least conserved residues (ConSurf score = 1) (*Ashkenazy et al., 2010*).

DOI: https://doi.org/10.7554/eLife.29795.009

**Figure supplement 2.** GST-Rac1 pull-down of different WRC mutants.

DOI: https://doi.org/10.7554/eLife.29795.010

observed binding in pull-down experiments, while the D Site, especially the surface patch formed by R961/P963/R964/Y967, has a major contribution. It is notable that P963 and Y967 are highly conserved in eukaryotic organisms including animals, fungi, amoebas and many protozoans, except plants (*Supplementary file 1*). In the experiments below, we focused on the Y967A mutation to disrupt the D Site.

Because Rac1 activates the WRC at membranes in vivo, we further developed an assay to examine if mutating the A or D Site could similarly disrupt the binding when Rac1 is attached to membranes (*Figure 3F*). We fused an N-terminal mCherry tag and a C-terminal His$_8$ tag to Rac1 (either Q61L/P29S or wild type, as indicated in *Figure 3F*). The latter allowed the protein to be anchored on supported bilayers containing 2% Ni$^{2+}$-NTA-DGS lipid (see schematic presentation in *Figure 3F*) (*Banjade and Rosen, 2014*). We then quantified the amount of EGFP-tagged WRC recruited to the Rac1-containing membrane using Total Internal Reflection Fluorescence (TIRF) microscopy. Paralleling the above GST pull-down experiments, mutating either the A Site (C179R) or the D Site (Y967A) significantly decreased membrane recruitment of the WRC, with the D Site mutation having a larger effect. In contrast, the S969F mutation did not decrease, but rather slightly increased membrane recruitment (*Figure 3F*).

These data suggest that Rac1 can bind to both the A and D sites. Nevertheless, it remained possible that only the D Site directly contacts the GTPase (as observed in the cryo-EM structure) and the A Site mutations disrupted binding through allosteric effects on the D site. To examine this possibility, we used Multi-Signal Sedimentation Velocity Analytical Ultracentrifugation (MSSV AUC) to directly measure the stoichiometry of the interaction between Rac1 (Q61L/P29S) and ΔWRC230 (*Figure 4*) (*Balbo et al., 2005*; *Padrick et al., 2010*). To separately track sedimentation of the two components, we fused EGFP to the N-terminus of Rac1. This labeling allowed us to use absorbance at 490 nm to specifically monitor EGFP-Rac1 and use interference signals to record the sedimentation of all proteins. Thus, in our assay EGFP-Rac1 was tracked by both signals, and the WRC by interference only. We note that MSSV AUC enables direct quantification of the EGFP-Rac1/WRC stoichiometry for any given species, as well as its sedimentation coefficient. Individually, both EGFP-Rac1 and the WRC sedimented as high-purity, monodisperse species, with sedimentation coefficients of 3.5 s and 10.5 s, respectively (*Figure 4*, and see *Figure 4—figure supplement 1* for SDS PAGE gels and gel filtration chromatography profiles of the protein samples). When the two proteins were mixed, keeping WRC at approximately 1 μM concentration and increasing EGFP-Rac1 from 1.2 μM to 15 μM, a fraction of EGFP-Rac1 began co-sedimenting with the WRC (*Figure 4A*, left panel, green curves). Addition of EGFP-Rac1 also shifted the WRC peak toward a higher sedimentation coefficient in a concentration dependent manner (*Figure 4A*, left panel, black curves). At an intermediate concentration of EGFP-Rac1 (3.7 μM), the WRC peak shifted to 11.7S with a Rac1:WRC stoichiometry of 1.3:1, suggesting more than one binding site. Increasing the concentration of EGFP-Rac1 up to 15.1 μM shifted the sedimentation coefficient of the WRC peak to 12.1S and stoichiometry to 1.9:1, confirming that two Rac1 molecules can simultaneously bind to a single WRC (*Figure 4B*).

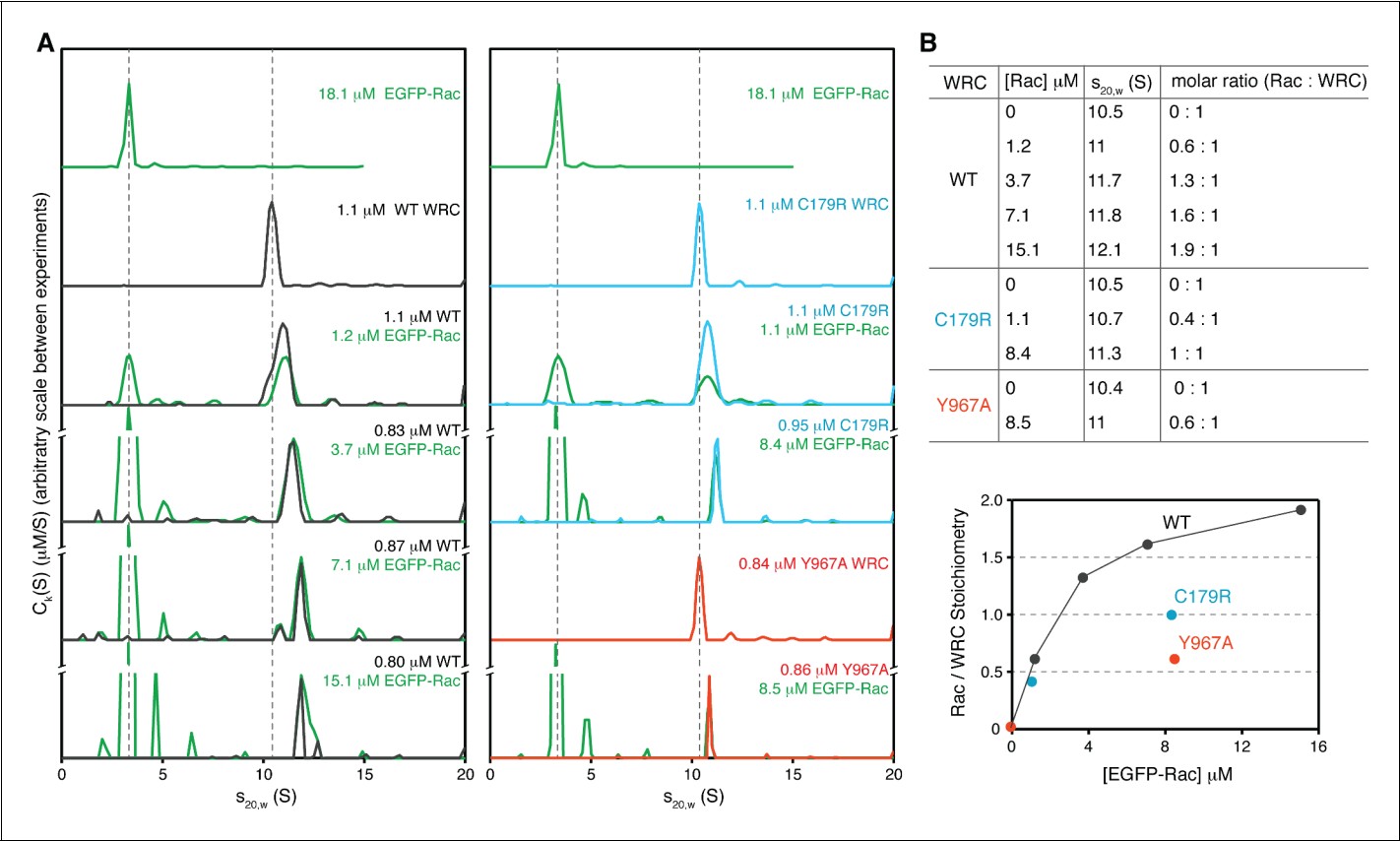

**Figure 4.** Stoichiometry of the Rac1/WRC complex determined by analytical ultracentrifugation. (**A**) Sedimentation profiles ($c_k(s)$ distributions) of indicated samples at various concentrations (green for EGFP-Rac1 (Q61L/P29S), black for WT ΔWRC230, blue for C179R, and red for Y967A). Dashed lines are used as a reference to indicate the sedimentation coefficients of the WRC alone and EGFP-Rac alone, respectively. (**B**) Sedimentation coefficient and stoichiometry of Rac1/WRC in the WRC peak.

DOI: https://doi.org/10.7554/eLife.29795.011

The following figure supplement is available for figure 4:

**Figure supplement 1.** Gel filtration chromatography profiles and Coomassie blue-stained SDS-PAGE gels of the ΔWRC230 and the EGFP-Rac1 samples used in the AUC experiments.

DOI: https://doi.org/10.7554/eLife.29795.012

The MSSV AUC data for the mutants were quite different. For the A Site mutant (C179R), at a nearly saturating concentration of EGFP-Rac1 (8.4 μM vs. dissociation constant $K_D$ of 0.27 μM as determined below), the binding stoichiometry only reached 1:1 (*Figure 4A*, blue curves, 4B). For the D Site mutant (Y967A), the binding stoichiometry only reached to 0.6:1 with 8.5 μM Rac1, consistent with the lower affinity ($K_D$ ~11.5 μM) of the A Site as determined below (*Figure 4A*, red curves, 4B) and implicated by the qualitative GST-pull down data above. Together, the AUC data confirmed that two molecules of Rac1 can engage the WRC at the A and D sites simultaneously, and that our mutations could selectively weaken binding to the individual sites.

The above data all suggest that the A Site has a weaker affinity than the D Site. To quantify this difference, we used an equilibrium pull-down (EPD) assay to determine the binding affinity of GST-Rac1 (Q61L/P29S) for each site (*Figure 5*) (*Lee et al., 2000*; *Pollard, 2010*). In this assay, we mixed a constant amount of ΔWRC230 (0.1 μM) with varying concentrations of GST-Rac1 (0.01 μM to 140 μM) and a fixed amount of Glutathione Sepharose beads. The beads were sufficient, given their quantity and the extremely high affinity of GST for glutathione ($K_D$ ~7 pM, [*Waterboer et al., 2005*]) to retain virtually all GST-Rac1 and its bound WRC, even at the highest concentrations (*Figure 5— figure supplement 1*). After a brief centrifugation to pellet the beads, uncomplexed WRC remained in the supernatant and could be quantified by SDS PAGE (example gels shown in *Figure 5A*),

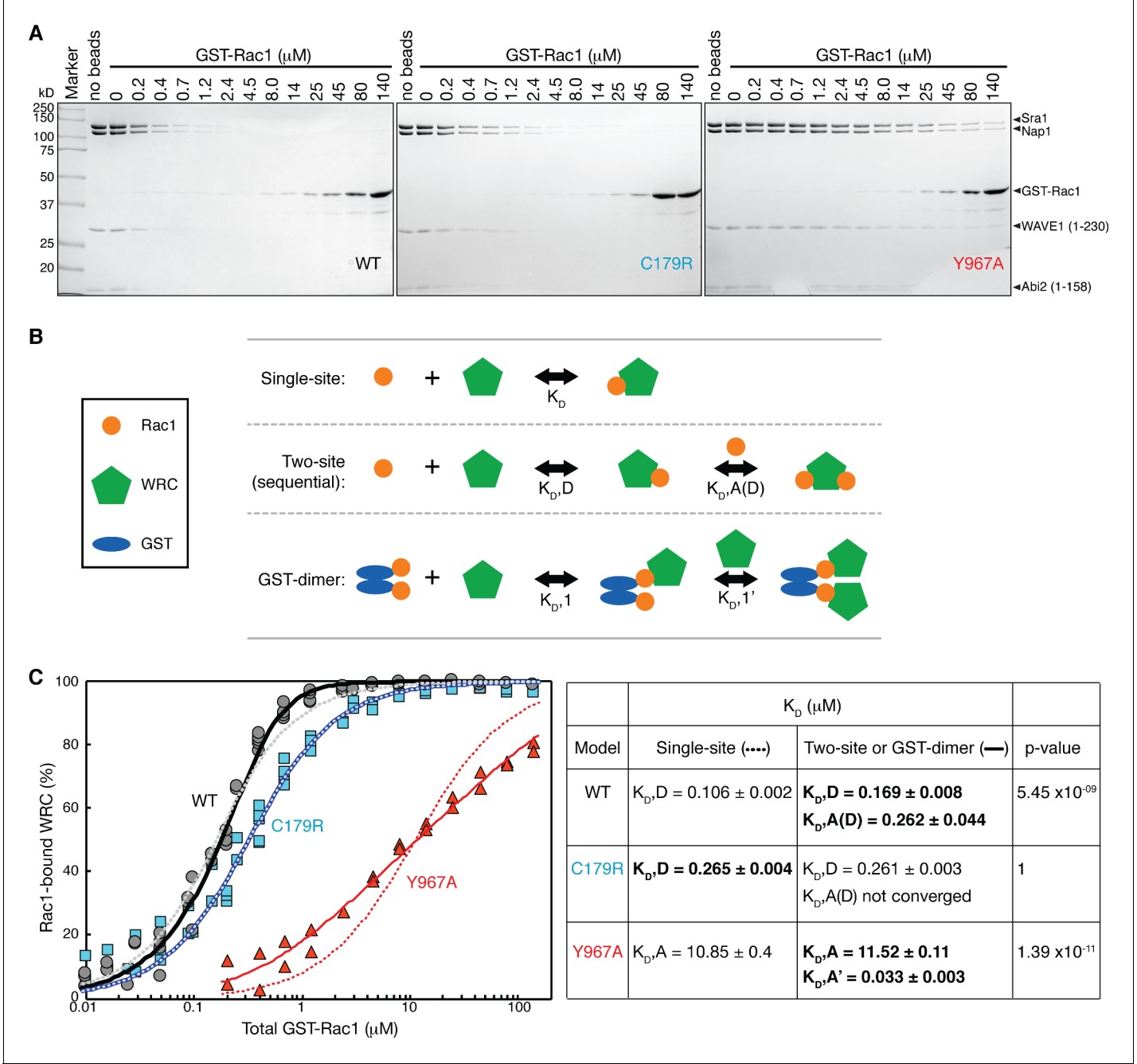

**Figure 5.** Quantification of the binding affinity by equilibrium pull-down assay. (**A**). Representative Coomassie-blue-stained SDS-PAGE gels of the supernatant samples in equilibrium pull down experiments involving GST-Rac1 (Q61L/P29S) binding to ΔWRC230. (**B**) Schematic representation of the three successful equilibrium models used to fit the binding isotherms. (**C**) Binding isotherms fit to the indicated models. For WT and C179R ΔWRC230, results for single-site (dotted curves) and two-site (solid curves) models are shown. For Y967A WRC, results for single-site (dotted curves) and GST-dimer models (solid curves) are shown. The data were pooled from multiple independent experiments (5 experiments for WT and C179R, and 2 experiments for Y967A) covering various concentration ranges. Derived dissociation constants are shown in the right table, with $K_D,D$ representing the dissociation constant for the D Site, $K_D,A$ for the A Site, $K_D,A(D)$ for the A Site when the D Site is occupied by Rac1, and $K_D,A'$ for the A Site of the second WRC in the GST-dimer model. Standard errors of the fit for the derived $K_D$ values are shown. *p* values were obtained by F-test; the most appropriate dissociation constants based on the *p* values are shown in bold. See *Figure 5—figure supplement 2* and Methods for additional details.

DOI: https://doi.org/10.7554/eLife.29795.013

The following figure supplements are available for figure 5:

**Figure supplement 1.** Representative Coomassie blue-stained SDS-PAGE gels of the supernatant (top gel) and the beads samples (bottom gel) in an equilibrium pull-down assay involving GST-Rac1 (Q61L/P29S) binding to ΔWRC230.

*Figure 5 continued on next page*

*Figure 5 continued*

DOI: https://doi.org/10.7554/eLife.29795.014

**Figure supplement 2.** Quantification of the binding affinity by equilibrium pull-down assay involving GST-Rac1 (Q61L/P29S) bindingΔWRC230.

DOI: https://doi.org/10.7554/eLife.29795.015

allowing us to determine the binding isotherms, which we could numerically fit to different binding models to obtain dissociation constants (*Figure 5B* and *Figure 5—figure supplement 2*). Note that the binding isotherms determined by the EPD assays do not distinguish the WRC species with one or two bound Rac1 molecules, or the WRC species with Rac1 bound to the D-Site or the A-Site, because all species can be equally pelleted with the glutathione beads. Therefore, fitting the binding isotherm of the WT WRC to a cyclic two-site binding model (*Figure 5—figure supplement 2A*) to extract the dissociation constants for both the D Site and the A Site could not converge, with large uncertainty for the derived $K_D$ (not shown). Fitting to a simplified sequential two-site binding model or to a single-site model resulted in a similar dissociation constant for the D Site ($K_D$,D of 0.169 ± 0.008 μM in *Figure 5C*, black solid curve compared to 0.106 ± 0.002 μM in *Figure 5C*, grey dotted curve). Fitting to the sequential two-site binding model was statistically better than the single-site model (p=5.45×10$^{-9}$), and yielded a dissociation constant for the A Site for the D Site-bound WRC ($K_D$,A(D)) of 0.262 ± 0.044 μM, suggesting positive cooperativity between the A and D Sites compared to the dissociation constant for the A Site alone ($K_D$,A of 11.52 ± 0.11 μM as determined below). Such cooperativity likely results from avidity, as two nearby Rac1 molecules, either partners in a GST fusion or adjacent on the solid matrix, bind the same WRC.

In contrast to the WT WRC, the isotherm for the A Site mutant (C179R), fit well to a single-site binding model, yielding a $K_D$,D of 0.265 ± 0.004 μM for the D Site (*Figure 5C*, light blue dotted curve), whereas fitting to a two-site binding model resulted in an identical $K_D$,D with no statistical improvement (*Figure 5C*, blue solid curve), supporting the idea that the C179R mutation effectively impaired the A Site.

For the D Site mutant (Y967A), the strong shift of the data to higher Rac1 concentrations clearly shows that the interaction that recruits the WRC to the beads is appreciably weaker than that for the wild type or A site mutant WRCs. But the isotherm did not fit well to either a single- or a two-site binding model, although both yielded a similar $K_D$,A for the A Site (10.85 ± 0.4 vs 9.6 ± 0.38 μM, *Figure 5C* red dotted curve, and see *Figure 5—figure supplement 2*). More complicated models involving more than two dissociation constants did not converge (not shown). However, when we modeled a dimeric GST-Rac1 binding two WRCs cooperatively (*Figure 5B*, GST-dimer model), the fitting was significantly improved (p=1.39×10$^{-11}$, *Figure 5C*, red solid curve). This model yielded a $K_D$,A of 11.52 ± 0.11 μM for binding of the first WRC, and a $K_D$,A' of 0.033 ± 0.003 μM for binding of the second WRC to this initial (GST-Rac1)$_2$-WRC assembly. These data would be consistent with strong positive cooperativity between WRC binding events when Rac1 only binds to the A Site. The better fit afforded by this model does not authenticate the molecular mechanism it describes. We note, however, that the potential for cooperative interactions between closely-positioned WRCs has been suggested by the crystal structure of miniWRC. There, the N-terminal helix of Sra1, which is not observed in the EM density of the dilute WRC studied here, bound between adjacent WRC assemblies in the crystal lattice (*Chen et al., 2010*). Future studies to examine such an interaction and its functional significance may be revealing. The GST-dimer model was not statistically better than the best fits for the WT or C179R WRC (*Figure 5—figure supplement 2B*; *Figure 5—figure supplement 2C*, thin solid curves).

In summary, our data demonstrate that both the previously identified A Site and the newly identified D Site are *bona fide* binding sites for Rac1, the latter with ~40 fold higher affinity than the former. The two sites can act together to recruit the WRC to membranes containing active Rac1. In retrospect, in our previous equilibrium dialysis experiments, the A Site mutants did not reach saturation at the highest concentrations of Rac1 used (8 μM), behavior we interpreted as indicating reduced affinity at a single site (see *Figure 3B* of [*Chen et al., 2010*]). Our new data provide a new interpretation for this observation, suggesting that the A Site mutants might have approached an asymptote at ~50% of the level reached by the WT WRC (i.e. with one site effectively eliminated by the mutation and a second site intact).

## Both Rac1-Binding sites contribute to WRC activation

We next asked whether both Rac1-binding sites are important for activating the WRC toward the Arp2/3 complex in pyrene-actin assembly assays (*Figure 6*). In these assays, actin assembly (monitored by an increase in fluorescence of pyrene-labeled actin) occurs slowly and with a long initial lag in the presence of WRC230VCA and Arp2/3 complex alone (orange curve in *Figure 6A*). Addition of Rac1 (Q61L)-GMPPNP activates the WRC, which, in turn, stimulates the actin-nucleating function of the Arp2/3 complex, decreasing the lag and increasing the maximum rate of actin assembly (dashed curves in *Figure 6A*).

For the WT WRC, a saturating concentration of Rac1 (10 µM) could produce actin-assembly rates equivalent to that of the constitutively active WAVE1 VCA peptide (dashed cyan curve in *Figure 6A*). Mutating the D Site (Y967A, *Figure 6B,E*) or the A Site (C179A, *Figure 6D,E*) completely abrogated activation, suggesting that activation of the WRC requires binding of Rac1 to both sites. Interestingly, the S969F mutant, which had slightly higher affinity for Rac1 (*Figure 3F*), was more sensitive to the GTPase than was WT WRC (*Figure 6A,C,E*). For example, the S969F mutant WRC was fully activated by only 3 µM Rac1, instead of 10 µM (green dashed curve in *Figure 6C* vs. cyan dashed curve in *Figure 6A*). Similarly, 1 µM Rac1 plus the S969F mutant WRC produced actin-assembly rates equivalent to 3 µM Rac1 plus the WT WRC (magenta dashed curve in *Figure 6C* vs. green dashed curve in *Figure 6A*). These data suggest that both Rac1-binding sites are important for activation of the WRC.

To further examine this notion, we re-analyzed the actin assembly data for Rac1 Q61L/P29S + WRC shown in *Figure 1—figure supplement 1C* in light of the dissociation constants derived from the EPD assays. As shown by the black data in *Figure 6F*, actin assembly rate is strongly non-linear with the total concentration of Rac1-bound WRC species, i.e. the sum of both WRC-$(Rac1)_1$ complexes (A or D site engaged) and WRC-$(Rac1)_2$. The lack of response to WRC-Rac1 concentrations up to 80 nM suggests suggests that engagement of a single site (primarily the D Site, due to its high affinity) is not sufficient to drive WRC activation. Further, plotting actin assembly rate versus the concentration of WRC-$(Rac1)_2$ yields a linear relationship (red data in *Figure 6F*). The slope of this line gives a specific activity of ~0.22 nM actin/s/nM WRC-$(Rac1)_2$. Together, the qualitative and quantitative actin assembly data show that engagement of both the A and D sites is necessary for WRC activation.

## Discussion

The Rho family GTPase, Rac1, plays important roles in directing signals from upstream pathways to the WRC to promote actin cytoskeletal assembly. We have determined the structure of a WRC-Rac1 complex using cryo-EM, revealing a GTPase-binding site distinct, and physically distant, from the site indicated previously by mutagenesis. Nevertheless, several lines of biochemical data demonstrate that both sites do, in fact, bind Rac1. Moreover, both are required for activation of the WRC toward the Arp2/3 complex. These findings have several important implications.

First, Rac1 activates the WRC through two binding sites that are distinct from the VCA-binding surface of Sra1. We previously posited that binding to the A Site might trigger conformational changes in the adjacent meander region. These, in turn, could destabilize the meander-VCA contacts, leading to release of the VCA from the body of the WRC (*Chen et al., 2010*). Such a mechanism, involving destabilization of a VCA-containing structural element, would be analogous to the activation of the WAVE1 relatives WASP and N-WASP by the Rho family GTPase Cdc42 (*Kim et al., 2000*; *Prehoda et al., 2000*; *Torres and Rosen, 2003*). Our current data suggest that the A Site and the D Site likely cooperate to trigger conformational changes that release the VCA. These changes may be focused in the meander region, as in our initial hypothesis, or may involve more substantial rearrangements of the WRC body, as suggested by the large distance between the D Site and the VCA-meander element of the assembly. In the latter scenario, the requirement for engagement of Rac1 at both sites to activate the WRC may explain why we did not observe major conformational changes on the WRC in our current structure, in which only the D Site is occupied. It remains to be understood how Rac1 binds the A Site and how the A and D Sites cooperate to drive allosteric activation of the WRC.

Second, our data inform on the recent observation that a point mutation, S968F, in Cyfip2 (corresponding to S969F in Sra1) caused phenotypes in mice including decreased response to cocaine and

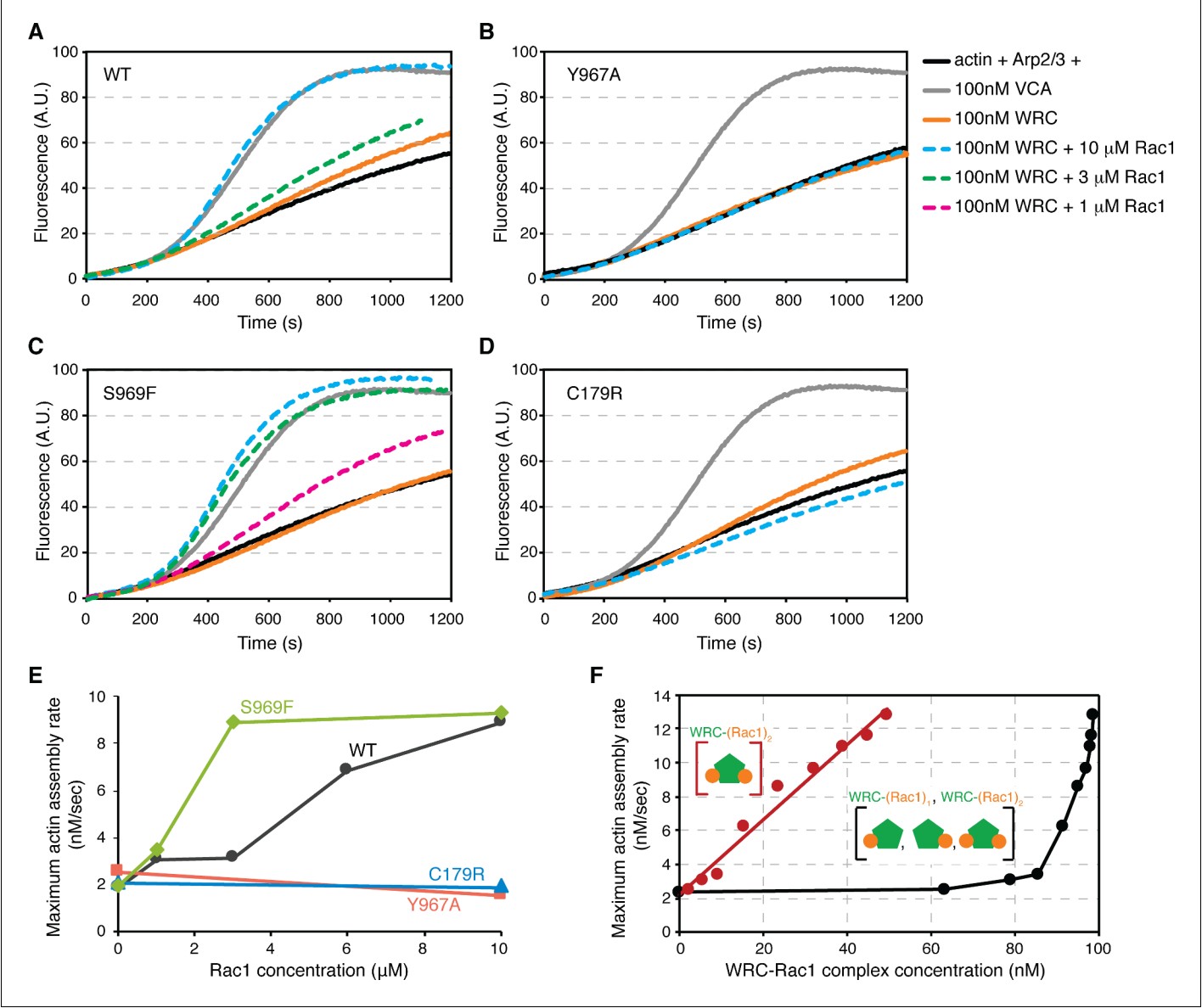

**Figure 6.** Activation of the WRC requires both Rac1-binding sites. (A–D) Actin assembly assays of the wild-type WRC230VCA (A), Sra1 Y967A (B), Sra1 S969F (C), and Sra1 C179R (D), with corresponding reaction conditions shown on the right. Reactions contain 4 µM actin (5% pyrene labeled), 10 nM Arp2/3 complex, 100 nM WRC230VCA or VCA, and indicated amount of Rac1 Q61L loaded with GMPPNP. (E) Maximum actin assembly rates at different Rac1 concentrations derived from data in (A–D) (*Doolittle et al., 2013*). (F) Maximum actin assembly rates at different concentrations of distinct WRC-Rac1 species, obtained from the data shown in *Figure 1—figure supplement 1C* for WRC activation by Rac1 Q61L/P29S. Concentrations of the indicated species of WRC-Rac1 complexes (shown as cartoons) formed at different Rac1 concentrations in the actin assembly assay were calculated using DynaFit (*Kuzmic, 1996*) by using a two-site cyclic model shown in *Figure 5—figure supplement 2A*, using $K_DD$ of 0.265 µM and $K_DA$ of 11.52 µM (derived from *Figure 5*, assuming no cooperativity between the A and D Site, since the actin assembly assays were performed in solution). Black data plot the summed concentration of all WRC-Rac1 species, while red data plot only WRC-(Rac1)$_2$. A linear fit of the latter data yields $r^2$ of 0.98.
DOI: https://doi.org/10.7554/eLife.29795.016

The following figure supplement is available for figure 6:

**Figure supplement 1.** Gel filtration chromatography profiles of the WRC230VCA samples used in the actin assembly assays.
DOI: https://doi.org/10.7554/eLife.29795.017

a reduced number of dendritic spines in neurons of the nucleus accumbens (*Kumar et al., 2013*). We found that S969 in Sra1 is buried immediately underneath the D Site. Unlike the previous hypothesis that this mutation may act by destabilizing the WRC, our results show that S969F in Sra1 facilitates Rac1 binding and WRC activation. It is possible that S969 may be part of the pathway for the conformational propagation from the D Site to the VCA-binding pocket upon Rac1 binding. Assuming the S968F mutation in Cyfip2 has a similar biochemical effect, our data suggest that excessive activation of the WRC by Rac1 might be responsible for the observed changes in cocaine response and neuronal morphology.

Third, our data depict how Rac1 could work together with other signaling molecules to regulate the WRC on membranes (*Figure 7*).One face of the WRC is highly basic, and the opposite face is highly acidic (*Chen et al., 2010*). This charge distribution suggests that when the WRC is bound to the negatively charged inner leaflet of the plasma membrane, and/or specific phospholipids such as PIP$_3$, its basic surface will lie against the membrane and its acidic surface will face the cytoplasm. In this orientation, both Rac1 binding sites can be positioned to enable a bound GTPase to bury its prenylated tail in the bilayer, consistent with an activation model involving simultaneous engagement of both sites. This orientation is also compatible with other potential interactions. For example, VCA can be readily released from above the membrane to interact with the Arp2/3 complex further in the cytoplasm (*Chazeau et al., 2014*). In addition, WIRS-containing transmembrane receptors could reach the WIRS-binding site on the acidic face of the WRC through < 23 residues (or ~80 Å in distance; *Figure 7A*) (*Chen et al., 2014a*). Indeed, with few exceptions, the WIRS elements in previously predicted WIRS-containing receptors in humans are generally located > 25 residues from their transmembrane sequence (not shown) (*Chen et al., 2014a*). These restraints together suggest a plausible model showing how the WRC may be oriented at membranes to interact with various regulatory molecules.

Finally, our bivalent interaction model provides an appealing mechanism to explain how cells could sense the level of active Rac1 at membranes to spatially and temporally control WRC activation and actin polymerization. Our data suggest that WRC activation requires simultaneous engagement of two Rac1 molecules at two distinct binding sites with ~40 fold different affinity. When the

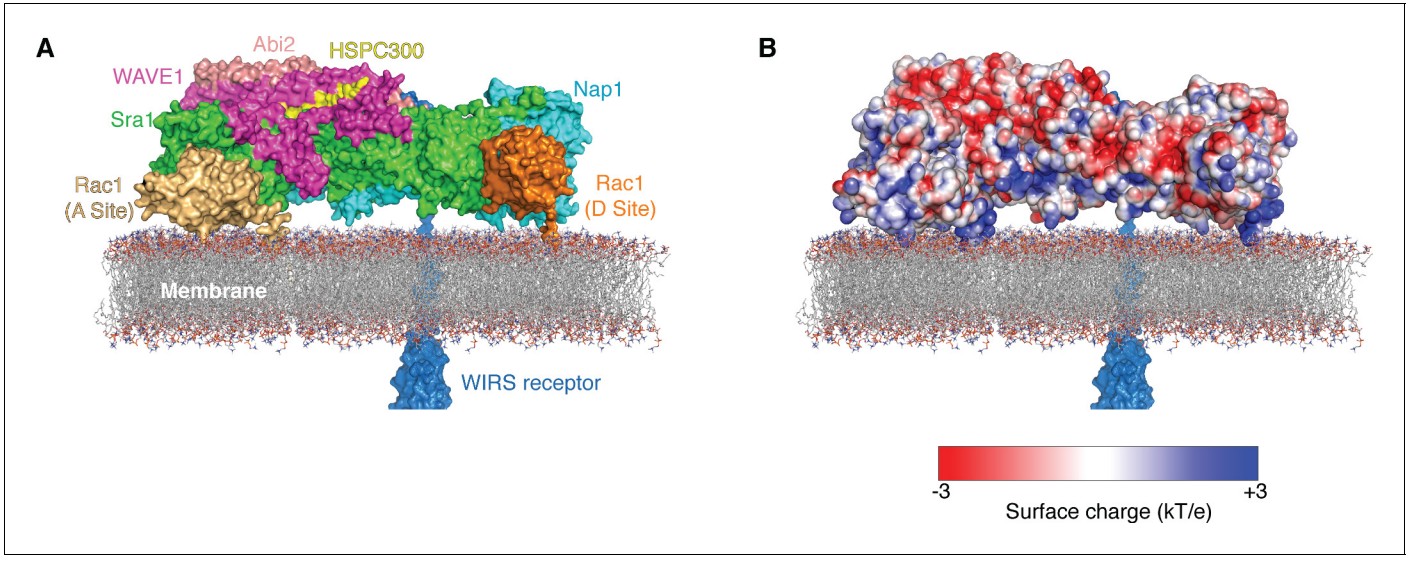

**Figure 7.** Two Rac1 molecules bind to the WRC on the membrane. Surface representation (**A**) and surface charge representation (**B**) (calculated using APBS in Pymol) (*Dolinsky et al., 2007*) show how the WRC can be oriented on the membrane to simultaneously interact with multiple signaling molecules, including two Rac1 molecules (gold and orange) anchored on the membrane through prenylation of their C-terminal tails, WIRS-containing surface receptors (dark blue), and acidic phospholipids through electrostatic interaction (represented by membrane). The Rac1 at the D site is taken from the ΔWRC230-Rac1 complex structure determined here; Rac1 at the A site was manually docked in a plausible orientation to contact Sra1 residues 179, 190, 434, 626 and 632 and place the C terminus near the membrane. WIRS-containing receptors could approach the WRC from the back (shown) or the front (not shown).

DOI: https://doi.org/10.7554/eLife.29795.018

density of Rac1 at membranes is low, Rac1 should interact with the WRC preferentially through the high-affinity site, which on its own is not sufficient to trigger activation but primes the complex. The WRC only becomes activated when Rac1 density is sufficient to enable binding to both sites. Consequently, activation of the WRC at membranes is likely non-linear with respect to Rac1 density (note that technical limitations in working with low Rac1 density on membranes prevented us from examining this experimentally). This behavior could be harnessed by cells to determine precisely where and when to turn on actin assembly. Analogous ideas have been suggested previously, based on observations that Rac1 and the Arf GTPase can cooperatively recruit and activate the WRC at membranes (*Koronakis et al., 2011*). Given the low sequence homology between Rac1 and Arf (22% and 38% identity in Switch I and Switch II, respectively), we feel it is unlikely that either the A or D site that we have characterized is also the binding site for Arf. Further, these cooperative effects could modulate the actions of other WRC ligands. For example, we previously discovered a large array of WIRS-containing receptors, which could potentially recruit the WRC to membranes (*Chen et al., 2014a*). For a given receptor, whether the recruitment would lead to productive activation of the WRC and actin polymerization would depend on the local membrane density of active Rac. Other regulators of the WRC, including kinases, acidic phospholipids and scaffolding proteins, might also cooperate with Rac1 activation in similar fashion (*Chen et al., 2010*; *Lebensohn and Kirschner, 2009*; *Miki et al., 2000*; *Padrick et al., 2011*). Moreover, the unstructured regions of WAVE and Abi (including the C terminal SH3 domain of Abi), which are not included in our current construct, may make additional contributions to Rac1 binding and activation. These regions have been shown to be important elements for regulating WRC activity, primarily by recruiting other regulatory molecules through interactions mediated by poly-proline sequences and/or SH3 domains (*Chen et al., 2014c*; *Dai and Pendergast, 1995*; *Miki et al., 2000*; *Soderling et al., 2002*) or by other mechanisms yet to be understood (*Miyamoto et al., 2008*). Together, these effects would lead to precise control of the timing and degree of actin assembly depending on the specific collection of upstream signals and their spatial organization at the membrane.

## Materials and methods

### Protein purification

Similar to mini-WRC, all WRC constructs used in this work contain human Sra1, Nap1, HSPC300 (or EGFP-HSPC300), and Abi2 (1-158), in addition to different WAVE1 constructs, including WAVE1(1–230)-$(GGS)_6$-Rac1(1-177)(Q61L/P29S) in $\Delta$WRC230-Rac1, WAVE1(1–217) in $\Delta$WRC217, WAVE1(1–230) in $\Delta$WRC230, WAVE1(1–217)-$(GGS)_6$-(485-559) in WRC217VCA, and WAVE1(1–230)-$(GGS)_6$-(485-559) in WRC230VCA. The WRCs were expressed and purified essentially as previously described for the mini-WRC (*Chen et al., 2010*; *Ismail et al., 2009*), except that the MBP-tagged WAVE1 proteins were expressed in ArcticExpress$^{TM}$ (DE3)RIL cells (Stratagene) at 10°C (*Chen et al., 2014b*).

Based on the following arguments the mutant WRCs all appear to be properly assembled and stable. (1) All mutations were made to surface residues, except the S967F mutation, which is buried immediately underneath the D Site (and nevertheless did not negatively affect Rac1 binding/activation). Such mutations likely only disrupt the local binding surface, rather than the overall folding of the WRC, especially since the WRC is ~400 kDa in size and is held together through multiple, extensive inter-subunit interactions. (2) Reconstitution of the recombinant WRC is a multi-step process, involving purification of individual proteins from different host cells and assembly/purification of sub-complexes and ultimately the WRC by a variety of affinity, ion exchange and gel filtration chromatography steps. All the mutant WRCs behaved identically to the wild type WRC during each step of the reconstitution (see example in *Figure 4—figure supplement 1* top panel, and *Figure 6—figure supplement 1*, comparing gel filtration chromatography profiles of mutant WRCs to the wild type). Furthermore, the C179R and Y967A mutants, each at a distinct Rac1 binding site, behaved identically to the wild type WRC in the MSSV AUC experiments. A mutation that disrupts the overall folding would very likely cause aberrant behaviors during certain steps in reconstitution, leading to lower purity and yield, or complete failure of reconstitution.

MBP-tagged mCherry-$(GGS)_6$-Rac1(1-188)-His$_8$ (mCherry-Rac1-His$_8$ for short) and MBP-tagged mEGFP-$(GGS)_6$-Rac1(1-188) (EGFP-Rac1 for short) were expressed in BL21 (DE3) T1$^R$ cells at 18°C

overnight and purified using amylose beads (New England Biolabs). After the MBP tag was removed by TEV-protease cleavage, the protein was further purified by Ni-NTA agarose beads (Qiagen, Germany), followed by cation-exchange chromatography through a Source SP15 column (GE Healthcare). All constructs were verified by DNA sequencing. Other proteins, including Arp2/3 complex, actin, WAVE1 VCA, GST-Rac1, Rac1 and TEV protease, were purified as previously described (*Ismail et al., 2009*).

## Electron microscopy

Prior to the EM experiments, an aliquot of the purified ΔWRC230-Rac1 complex was thawed and passed through a 2.4 mL Superdex 200 gel-filtration column (GE Healthcare) to remove glycerol from previous purification steps and exchange buffer to 10 mM HEPES pH 7.0, 100 mM NaCl, 1 mM $MgCl_2$ and 2 mM TCEP. The protein sample (3.5 μl at 0.1 mg/mL) was applied to glow-discharged Quantifoil holey carbon grids (400 copper mesh, R1.2/1.3), which were then flash frozen in liquid ethane using a Gatan CryoPlunge 3. Grids were imaged with an FEI Tecnai F20 operated at an acceleration voltage of 200 kV and a calibrated magnification of 40,410x (nominal magnification of 29,000x), yielding a pixel size of 0.62 Å on the specimen level. A Gatan K2 Summit direct detector device (DDD) camera was used to collect dose-fractionated image stacks in super-resolution counting mode using the UCSF *Figure 4* data acquisition software (*Li et al., 2013*). The dose rate used was 6.4 $e^-/Å^2/s$. Of a total of 2173 image stacks, 30 frames were recorded with 200 ms per frame (total exposure time of 6 s) for 1366 stacks, 34 frames were recorded with 300 ms per frame (total exposure time of 10.2 s) for 407 stacks, and 51 frames were recorded with 200 ms per frame (total exposure time of 10.2 s) for 400 stacks. The frames were binned over 2 × 2 pixels (yielding a pixel size of 1.24 Å), aligned to each other using motioncorr (*Li et al., 2013*), and summed.

## Image processing

The defocus parameters of each micrograph were determined with CTFFIND3 (*Mindell and Grigorieff, 2003*). Particles from the first 489 images were manually picked using e2boxer (*Tang et al., 2007*) and windowed into 220 × 220 pixel images, which were then reduced to 64 × 64 pixel images. The 11,874 particle images were subjected to the iterative stable alignment and clustering (ISAC) procedure (*Yang et al., 2012*) implemented in SPARX (*Hohn et al., 2007*). Three ISAC generations, specifying 200 particles per group and a pixel error threshold of 0.7, resulted in classification of 2607 particles (~22% of the entire data set) into 56 classes.

All images were then automatically picked with Relion1.3 (*Scheres, 2012*), yielding 160,591 particles that were windowed into 250 × 250 pixel images. The particle images were subjected to 2D classification specifying 300 classes, and classes producing poor averages were discarded. This step was repeated specifying 200 classes. The remaining 78,406 particles were subjected to 3D classification specifying 12 classes and using as initial model the crystal structure of the mini-WRC (PDB 3P8C) (*Chen et al., 2010*) low-pass filtered to 60 Å. Six classes showing extra density for Rac1 (39,145 particles) were combined and subjected to a second round of 3D classification specifying 10 classes. Of the resulting maps, 8 classes showed extra density for Rac1 (*Figure 1—figure supplement 2D*) and were combined (29,784 particles) for refinement, yielding a map at a resolution of 7.4 Å.

For particle polishing in Relion (*Scheres, 2012*), image stacks were created for which each frame represented 600 ms of exposure. For instance, for image stacks recorded with 200 ms frames, 3 frames were averaged; for image stacks recorded with 300 ms frames, 2 frames were averaged. Only the first 10 frames, corresponding to an exposure time of 6 s, were used for particle polishing. The final map had a resolution of 7.0 Å, as estimated by Fourier shell correlation of independently refined 3D reconstructions from half data sets using the 0.143 cut-off criterion (*Figure 1—figure supplement 3B*). The local resolution was assessed using the program ResMap (*Kucukelbir et al., 2014*) (*Figure 1—figure supplement 3C*). Atomic models of the mini-WRC (PDB 3P8C) and Rac1 (PDB 3SBD) were docked into the EM map using UCSF Chimera (*Pettersen et al., 2004*). Figures were prepared with UCSF Chimera and Pymol (*Schrodinger, 2015*). The EM density map has been deposited in EMDB with accession number EMD-6642.

## Non-equilibrium GST Pull-down assay

Non-equilibrium GST pull-down experiments in *Figure 3E* were performed as previously described (*Chen et al., 2014a*). Briefly, 130 pmol of GST-Rac1(1-177)-GMPPNP and 260 pmol of WRC (composed of Sra1, Nap1, MBP-EGFP-HSPC300, MBP-Abi2(1-158) and MBP-WAVE1(1–230)) were mixed with 20 µL of Glutathione Sepharose beads (GE Healthcare) in 1 mL of binding buffer (20 mM HEPES pH 7, 50 or 100 mM NaCl, 5% (w/v) glycerol, 2 mM $MgCl_2$ and 5 mM β-mercaptoethanol) at 4°C for 30 min, followed by three washes using 1 mL of the binding buffer. Bound proteins were eluted with GST elution buffer (100 mM Tris-HCl pH 8.5, 50 mM NaCl, 5% (w/v) glycerol, 2 mM $MgCl_2$, 5 mM β-mercaptoethanol and 30 mM reduced glutathione) and examined by SDS-PAGE. *Figure 3—figure supplement 2A and B* used GST-Rac1(1-177)-GDP and GST-Rac1(1-177) (Q61L/P29S), respectively and the same WRC assemblies as *Figure 3E*.

## Equilibrium GST pull-down assay

Equilibrium GST pull-down (EPD) experiments were performed essentially as previously described (*Lee et al., 2000*; *Pollard, 2010*). Glutathione Sepharose beads (GE Healthcare) were first equilibrated in EPD buffer (20 mM HEPES pH 7, 100 mM NaCl, 5% (w/v) glycerol, 2 mM $MgCl_2$, and 5 mM β-mercaptoethanol) and stored as a 50% (v/v) slurry. All protein samples were dialyzed against EPD buffer overnight at 4°C to maximize buffer match. After dialysis, GST-Rac1(1-188) (Q61L/P29S) was concentrated to ~500 µM using an Amicon Ultra centrifuge concentrator (3 kDa MWCO, Millipore). All proteins were centrifuged at ~21,000 g at 4°C for 10 min to remove denatured proteins before use. Each reaction was assembled in 100 µL total volume of EPD buffer in a 200 µL PCR tube (Axygen), which contained 0.1 µM ΔWRC230 (consisting of Sra1, Nap1, HSPC300, MBP-Abi2(1-158) and MBP-WAVE1(1–230)) varying concentrations of GST-Rac1(1-188) (Q61L/P29S), 30 µL of the Glutathione Sepharose beads (by aliquoting 60 µL of the 50% (v/v) slurry using a wide-bore pipette tip), and 0.05% Triton X100 to facilitate mixing. The reactions were gently mixed at 4°C on a rotary mixer for 30 min. After a brief centrifugation (~10,000 g for 30 s) to pellet the beads, 40 µL of the supernatant was immediately transferred to 8 µL of 6X loading buffer (360 mM Tris-HCl pH 6.8, 12% (w/v) SDS, 60% (w/v) glycerol, 0.00012% (w/v) bromophenol blue, and 140 mM freshly added 2-mercaptoethanol), and analyzed by Coomassie blue-stained SDS-PAGE gels. The gels were imaged by a ChemiDoc[TM]XRS + system (BioRAD) using its standard protocols. Total intensity of the Sra1 and Nap1 bands was quantified by ImageJ (FIJI) to determine the unbound WRC. The derived fractional occupancy from 2 to 5 independent experiments was directly merged to obtain the binding isotherms. The program DynaFit (BioKin [*Kuzmic, 1996*]) was used to numerically fit the binding isotherms to different equilibrium models to obtain dissociation constants $K_D$ (see *Supplementary file 2* for scripts). The uncertainty of the derived $K_D$s was further evaluated by Monte Carlo simulations with a 'shuffle' algorithm implemented in DynaFit. The final $K_D$ values reported in *Figure 5* together with the standard errors of the fit were determined by the histograms generated by 5,000 Monte Carlo simulations (*Kuzmic, 1996*). The fitting results were compared by the F-test using Matlab (Mathworks).

## Membrane recruitment assay

Supported lipid bilayers (SLBs) containing 98% POPC and 2% $Ni^{2+}$-NTA DGS (Avanti Polar lipids) were prepared as previously described (*Banjade and Rosen, 2014*; *Su et al., 2016*), with the following modifications. After washing with 5% Hellmanex III (Hëlma Analytics), glass bottom 96-well plates (Matrical) were washed with 6 M NaOH for 30 min at 45°C twice, and thoroughly rinsed with MilliQ $H_2O$. Small unilamellar vesicles (SUVs) were added to cleaned wells equilibrated with 50 mM HEPES pH 7.5, 150 mM NaCl and 1 mM TCEP, and incubated at 40°C for 1 hr.

The membrane recruitment assay was performed at room temperature. The SLBs were first washed twice with binding buffer containing 10 mM HEPES pH 7.0, 100 mM NaCl, 2 mM $MgCl_2$, 1 mM TCEP and 1 mg/mL BSA. mCherry-Rac1-His[8] protein (either Q61L/P29S or wild type, as indicated in *Figure 3F*) was added at 20 nM concentration, and allowed to bind the lipid bilayers for 30 min. Unbound proteins were then removed by three washes, which together afforded a 125-fold dilution of the initial solution. EGFP-tagged WRCs (composed of Sra1, Nap1, MBP-EGFP-HSPC300, MBP-Abi2(1-158) and MBP-WAVE1(1–230)) were then added at 80 nM and incubated for 30 min,

followed by three washes affording a 125-fold dilution of the initial solution to reduce non-specific binding of the WRC as well as background fluorescence.

TIRF images were taken using a TIRF/iLas2 module (Biovision) mounted on a Leica DMI6000 microscope with a 100 × 1.49 NA objective (EM-CCD camera, ImagEMX2, Hamamatsu). Images were acquired at 488 nm for EGFP-WRC and 561 nm for mCherry-Rac1. Images were processed using ImageJ (FIJI). Signal from the 488 nm channel from samples with only mCherry Rac1 and binding buffer was used as background for EGFP-WRC measurements. Signal from the 561 nm channel from samples with only SLBs and EGFP-WRC was used as background for mCherry-Rac1 measurements. EGFP/mCherry signals were calculated from 10 images from two independent experiments. Data are reported as mean ± SEM. The autofocus module on our microscope was used to find the focal plane for all measurements.

## Analytical Ultracentrifugation

The Multi-Signal Sedimentation Velocity (MSSV) Analytical Ultracentrifugation (AUC) assay was performed as previously described (*Balbo et al., 2005*; *Padrick et al., 2010*), to measure the stoichiometry of binding between EGFP-Rac1 (1-188) (Q61L/S29S) and ΔWRC230 (consisting of Sra1, Nap1, HSPC300, MBP-Abi2(1-158) and MBP-WAVE1(1–230)). Extra care was given when preparing the protein samples to ensure good data quality since the WRC is observed to be sensitive to denaturation, likely caused by liquid surface tension from air bubbles and mechanical shaking, especially in the absence of glycerol in the buffer. The sample preparation required gentle pipetting, avoiding air bubbles or foaming, and removing sticky, denatured WRC floating on the surface after centrifugation steps prior to AUC analyses (same cautions were applied to all other experiments reported here). Both the WRC and EGFP-Rac1 samples were passed through a Hiload Superdex 200 gel filtration column (GE Healthcare) equilibrated with 20 mM HEPES pH 7, 150 mM NaCl, 10 or 20% (w/v) glycerol and 1 mM DTT (and 2 mM $MgCl_2$ for EGFP-Rac1) to remove potential aggregates. The proteins were aliquoted, flash frozen in liquid nitrogen, and stored at −80°C. Before use, proteins were thawed in a water bath at room temperature and subjected to extensive dialysis (for 3 continuous days with multiple buffer exchanges at 4°C) in the same beaker containing AUC buffer (10 mM HEPES pH 7, 100 mM NaCl, 2 mM $MgCl_2$, and 1 mM DTT) to remove glycerol (which could affect the AUC interference signal) and to maximize buffer match. After dialysis, the samples were centrifuged at ∼21,000 g for 10 min at 4°C to remove denatured proteins. Approximately 400 µL of the samples were introduced into the sample sectors of dual-sector Epon centerpieces that had been placed between sapphire windows in a standard AUC cell. The reference sectors were filled with the same volumes of AUC buffer. The cells were inserted into an An50Ti rotor and put under vacuum for temperature equilibration for a minimum of 2 hr. Subsequently, centrifugation was commenced at 50,000 rpm. Data were acquired using both absorbance at 490 nm and interference optics. All experiments were performed overnight at 20°C. Interferometric molar signal increments for the two protein species were calculated based on their respective amino-acid compositions (*Zhao et al., 2011*), resulting in 141,801 fringes $M^{-1}cm^{-1}$ for EGFP-Rac1 and 904,251 fringes $M^{-1}cm^{-1}$ for the WRC. Using the former value as a standard, the data sets for EGFP-Rac1 alone were globally analyzed to establish its sedimentation coefficient and the extinction coefficient (*Padrick et al., 2010*) at 490 nm (32,310.8 AU $M^{-1}cm^{-1}$. This value is lower than the reported value of 56,000 AU $M^{-1}cm^{-1}$ (*Cranfill et al., 2016*), likely due to incomplete maturation of the EGFP tag during overnight expression (*Heim et al., 1995*). A data set for WRC alone was analyzed using only interference data to determine its sedimentation coefficient. The actual concentrations of each protein species in the assembled cells were calculated by integrating the $c_k(s)$ distributions as described below.

The MSSV data were analyzed using SEDPHAT (*Balbo et al., 2005*). Using the extinction coefficients obtained above, the interferometric and absorbance (at 490 nm) data from mixtures of the WRC and EGFP-Rac1 were globally analyzed to yield component $c_k(s)$ distributions that reported on the concentrations of WRC and EGFP-Rac1 as a function of sedimentation coefficient. By integrating these distributions over sedimentation coefficient ranges where co-sedimentation was evident, the concentrations of the individual proteins in the complex could be calculated, and thus the molar ratios of the complexed WRC and EGFP-Rac1 were derived. Sedimentation coefficient values for the individual proteins and complexes were determined by a weighted integration scheme (*Schuck, 2003*), yielding signal-average sedimentation coefficients. In combination with the

calculated molar ratios and refined frictional ratios (*Padrick et al., 2010*), these values allow the determination of the stoichiometries of the complexes.

## Pyrene-actin assembly assay

Actin polymerization assays were performed at 22°C using a PTI Fluorometer (Photon Technology International) as previously described (*Chen et al., 2014a*). Reactions contained 4 µM actin (5% pyrene labeled), 10 nM Arp2/3 complex, 100 nM WRC217VCA (*Figure 1—figure supplement 1c*; consisting of Sra1, Nap1, HSPC300, Abi2(1-158) and WAVE1(1–217)-(GGS)$_6$-(485-559)) or WRC230VCA (*Figure 6*; consisting of Sra1, Nap1, HSPC300, Abi2(1-158) and WAVE1(1–230)-(GGS)$_6$-(485-559)) or VCA, and various amounts of Rac1 Q61L loaded with GMPPNP.

## Acknowledgements

We thank Shae Padrick for discussions regarding model fitting, and Scott Nelson for discussion and guidance in using the DynaFit program. We thank an anonymous referee for suggesting the equilibrium pull-down experiment to quantify the affinity of GST-Rac1 for the WRC. Research was supported by grants from the NIH and Welch Foundation to MKR (R01-GM056322 and I-1544, respectively), the Howard Hughes Medical Institute, and start-up funds to BC from the Iowa State University and the Roy J Carver Charitable Trust.

## Additional information

### Funding

| Funder | Grant reference number | Author |
| --- | --- | --- |
| National Institutes of Health | R01-056322 | Michael K Rosen |
| Welch Foundation | I-1544 | Michael K Rosen |
| Howard Hughes Medical Institute | | Michael K Rosen |
| Iowa State University & Roy J. Carver Charitable Trust | | Baoyu Chen |

The funders had no role in study design, data collection and interpretation, or the decision to submit the work for publication.

### Author contributions

Baoyu Chen, Conceptualization, Data curation, Formal analysis, Supervision, Funding acquisition, Project administration, Writing—review and editing; Hui-Ting Chou, Conceptualization, Data curation, Formal analysis, Investigation, Methodology, Writing—original draft, Writing—review and editing; Chad A Brautigam, Wenmin Xing, Lisa Henry, Data curation, Formal analysis, Investigation, Methodology, Writing—review and editing; Sheng Yang, Independently replicated the GTP- and GDP-Rac1 pull down experiments shown in Figure 3; Lynda K Doolittle, Thomas Walz, Resources; Michael K Rosen, Data curation, Formal analysis, Supervision, Funding acquisition, Project administration, Writing—review and editing

### Author ORCIDs

Baoyu Chen (iD) http://orcid.org/0000-0002-6366-159X
Michael K Rosen (iD) http://orcid.org/0000-0002-0775-7917

### Decision letter and Author response

Decision letter https://doi.org/10.7554/eLife.29795.022
Author response https://doi.org/10.7554/eLife.29795.023

## Additional files

**Supplementary files**

• Supplementary file 1. Sequence alignments of Sra1 proteins from representative organisms. Residues in the A Site mutated previously are colored grey. Surface residues shown in *Figure 2A–B* are highlighted using the same color scheme. For clarity, insertions in three of the sequences were hidden and indicated by amino acids colored in cyan.
DOI: https://doi.org/10.7554/eLife.29795.019

• Supplementary file 2. Scripts used in the DynaFit program to fit the equilibrium pull-down data.
DOI: https://doi.org/10.7554/eLife.29795.020

• Transparent reporting form
DOI: https://doi.org/10.7554/eLife.29795.021

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
