## [Decision Letter]

Thank you for submitting your article "Rac1 GTPase Activates the WAVE Regulatory Complex Through Two Distinct Binding Sites" for consideration by *eLife*. Your article has been reviewed by two peer reviewers, and the evaluation has been overseen by a Reviewing Editor and Tony Hunter as the Senior Editor.

The Reviewing Editor has drafted this decision to help you prepare a revised submission.

Our view is that this is a well-executed study that provides novel information about the regulation of actin assembly and membrane dynamics. In a previous round of review, there was a question around mutagenesis: did the authors make any measurements to verify that the mutant proteins are stable and well folded? This control would verify that the mutations are interfering with binding rather than protein structure.

The authors responded with good arguments that the mutated proteins are folded. Doing a simple urea-denaturation experiment while monitoring the intrinsic fluorescence of the complex might have revealed some loss of stability, which is different from being folded. Examples exist of apparently innocuous surface mutations that change the stability a protein.

We ask the authors to present their arguments why the mutants are properly folded in the manuscript and show gel filtration profiles for the most important mutants. Moreover, input lanes for the most important mutants along with wild type must be shown in the pull-down gels.

---

## [Author Response]

We ask the authors to present their arguments why the mutants are properly folded in the manuscript and show gel filtration profiles for the most important mutants.

We have now included in the Materials and methods section a paragraph containing our arguments that the mutant WRCs are most likely properly folded and assembled, with a call-out in the main text (second paragraph in subsection “Rac1 Binds to Two Distinct Sites on the WRC with Different Affinities”). We have also added gel filtration chromatography traces for three additional mutants in a new Figure 6—figure supplement 1.

Moreover, input lanes for the most important mutants along with wild type must be shown in the pull-down gels.

We have moved the input gel for the pull down experiments into the main Figure 3, lower panel. The pull down data shown in Figure 3 and Figure 3—figure supplement 3 were performed in parallel, so the same input gel applies to all panels. This is now stated in the legend to Figure 3—figure supplement 3.

We have also gone through the manuscript carefully to make sure that all Rac and WRC constructs used in each experiment are clearly indicated in the text, methods and figure legends.